# Fossil birds from the Roof of the World: The first avian fauna from High Asia and its implications for late Quaternary environments in Eastern Pamir

**Nikita Zelenkov**[1]\*, **Nuritdin Sayfulloev**[2], **Svetlana V. Shnaider**[3]

**1** Borissiak Paleontological institute of Russian Academy of Sciences, Moscow, Russia, **2** Institute of History, Archaeology and Ethnography, Dushanbe, Tajikistan, **3** ArchaeoZOOlogy in Siberia and Central Asia–ZooSCAn, CNRS–IAET SB RAS International Research Laboratory, IRL 2013, Institute of Archaeology and Ethnography, Siberian Branch of the Russian Academy of Sciences, Novosibirsk, Russia

\* nzelen@paleo.ru

**Data Availability Statement:** All relevant data are within the manuscript.

## Abstract

The Eastern Pamir (eastern Tajikistan) is a high-mountain plateau with elevations up to 7000 m, currently characterized by extremely severe environmental conditions and harboring a specialized montane fauna, which in part is shared with that of the Tibetan Plateau. The modern bird fauna of High Asia comprises a diversity of both ancient and recently diverged endemics, and thus is of general importance for historical biogeography and understanding the origin of modern high mountain ecosystems. However, the past history of the Central Asian highland avian communities remains practically unknown, as no fossil bird assemblages from high elevation areas were previously reported. In particular, it remains completely unexplored how birds responded to late Quaternary climatic fluctuations. Here we report the first fossil bird fauna from the High Asia and the first fossil birds from Tajikistan. An assemblage from the late Pleistocene through middle Holocene of Istykskaya cave (4060 m) in Eastern Pamir surprisingly comprises a remarkable diversity of waterbirds, including a few species that are completely absent from High Asia today and that were not reported globally from such high altitudes. The diversity of waterbirds incudes taxa of various ecological preferences (shorebirds, underwater and surface feeders, both zoophagous and phytophagous), strongly indicating the presence of a productive waterbody at the vicinity of the site in the past. These observations correspond to recent palaeoclimatic data, implying increased water availability in this region, currently occupied by high mountain semi-deserts. Our findings for the first time show that milder environmental conditions of late Quaternary attracted lowland species to the Central Asian highland wetlands. The reported assemblage yet contains several characteristic highland taxa, indicating a long-time persistence of some Central Asian montane faunistic elements. In particular, it includes the Tibetan Sandgrouse (*Syrrhaptes tibetanus*), a highly-specialized montane dweller, which is for the first time found in the fossil record.

**Funding:** The fieldwork was supported by the Russian Foundation for Basic Research (RFBR) 18-09-40081 «Initial human colonisation in the highlands of Western Central Asian (Pamir and Alay valley): cultural dynamics, chronology, palaeogeography» (https://www.rfbr.ru). The radiocarbon dating was supported by the Russian Science foundation (RSF) 19-78-10053 «The emergence of food-producing economies in the high mountains of interior Central Asia» (https://rscf.ru). Laboratory investigation of bird remains was supported by grant from RFBR (project 20-04-00975). The funders had no role in study design, data collection and analysis, decision to publish, or preparation of the manuscript.

**Competing interests:** The authors have declared that no competing interests exist.

## Introduction

Highlands cover vast territories in Central and southern Asia, and are characterized by distinctive avian faunas containing numerous endemic species [1–3]. Although the evolutionary history of the Central Asian mountain avian faunas is of special interest, direct paleontological evidence for the past history of highland birds is extremely limited. From high elevations (~2000 to 2700 m above sea level) of the Tibetan Plateau only a couple of fossil birds were previously reported [4, 5]. There are no published data on the past avian diversity of the more westerly located mountain ranges, such as Pamir, Karakoram or Tian Shan. Furthermore, no fossil bird faunas were previously reported from highlands of Central Asian Mountains in general. But mountain ranges and plateaus are important areas harboring a remarkable diversity of both relictual and recently diverged taxa and hence playing a significant role in the global biogeography [1]. Additionally, birds are sensitive environmental indicators [6–11], and thus provide important source information for palaeogeographic and palaeoecological reconstructions.

Here we describe an assemblage of bird bones, dated to late Pleistocene through middle Holocene [12, 13], from a cave in southeastern Pamir Mountains (eastern Tajikistan). The Eastern Pamir is the western part of High Asia, metaphorically described as the Roof of the World, and it is a highland plateau with elevations of 3500–7000 m, presently covered by cold mountain steppes and deserts. This region is characterized by a very harsh climate with extremely poor annual precipitation (below 40 mm for the particular area [14]), low mean annual temperatures (at least -2.5C) and prolonged period of snow cover (at least about 240 days annually; [15]). The site (Istykskaya cave) is located 4060 meters above sea level [12], and thus the reported assemblage is the first and the only known high altitude avian fauna in Asia. The avian remains from Istykskaya cave for the first time yields insights into the past history of birds of Pamir Plateau and indicate environment conditions, which attracted a diversity of water bird species, including lowland taxa that are no longer present in this region.

## Material and methods

The Istykskaya cave (37˚44'05.9", 074˚22'15.8") is located in the SE part of Eastern Pamir (eastern Tajikistan) at the absolute elevation of 4060 m a.s.l. (Fig 1A–1C). The cave is located on the left bank of Sul-Istyk river (left tributary of the Murghab river) some 65 km southeast of Murghob town. The cave was first explored in 1970s by V.A. Zhukov, who excavated a total area of 30 m$^2$ [13]. Most recently in 2018, a field team led by NS and SVS cleaned the northeastern section of the original Zhukov's excavations in the drop line of the cave, and further excavated an area of 2x2 m$^2$ in 2019. Four stratigraphic units were identified during archaeological work (Fig 1D and Table 1).

Field works at the site were carried out under the terms of an international scientific cooperation agreement between the Institute of History, Archaeology and Ethnography of the Republic of Tajikistan and the Institute of Archaeology and Ethnography of SB RAN of Russia (2017–2021), and were licensed by the Ministry of Culture of the Republic of Tajikistan.

We conducted a new absolute dating of Istykskaya cave assemblage using samples obtained from two bone fragments found within archaeological collections from excavations in 2018 stored at the Archaeological Department of Institute of History, Archaeology and Ethnography (Dushanbe, Tajikistan). The samples were processed and dated at the Golden Valley laboratory of Novosibirsk Center for Cenozoic Geochronology (Russia) giving a calibrated date of 4410–3981 for cultural layer 1 (GV 02109) and 8635–8192 for cultural layer 2 (GV 02111). Given date ranges correspond to 2-sigma (95.4%) probability range, calibrated using IntCal20

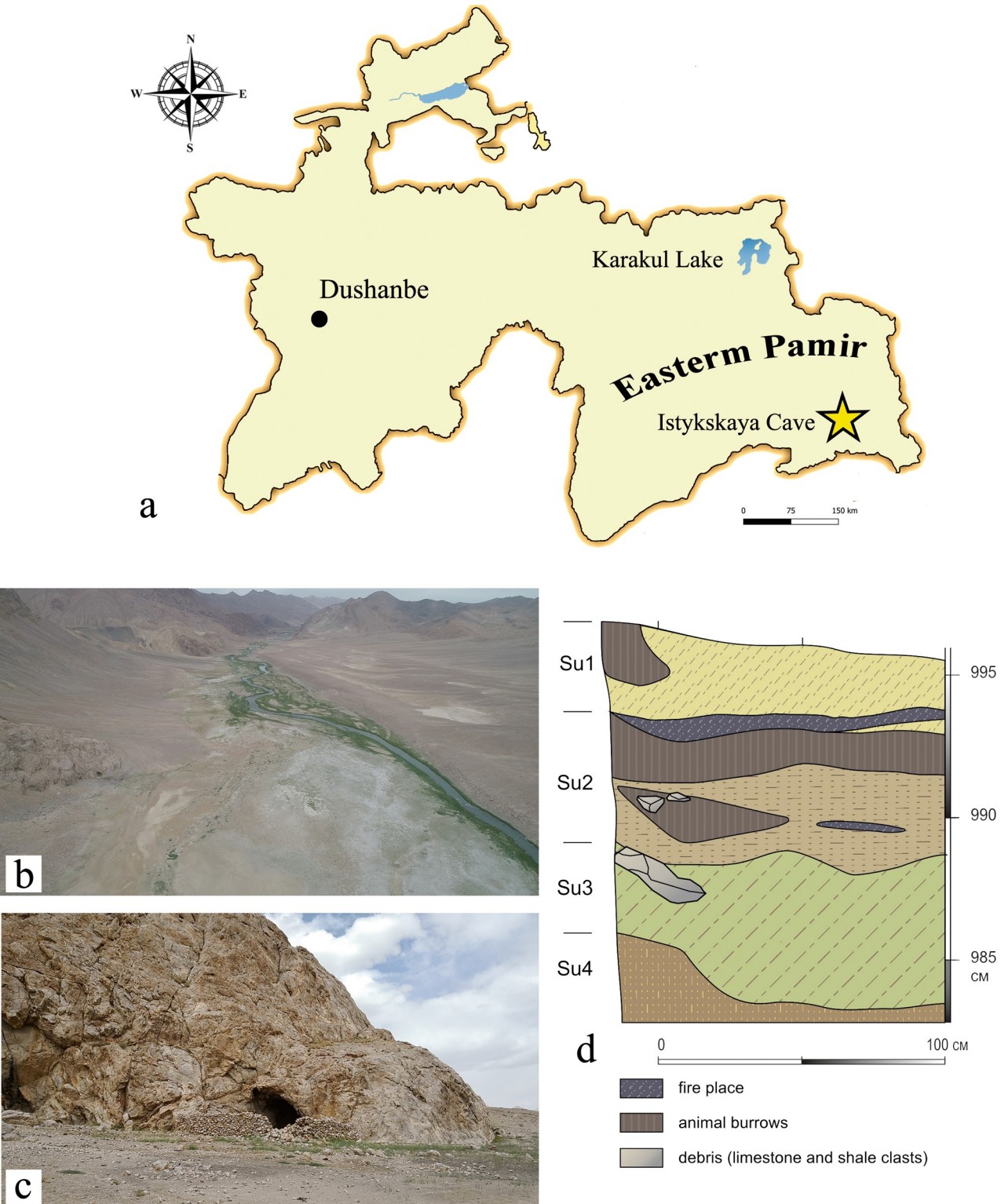

**Fig 1. Istykskaya cave.** (a) Site location (star) in the Pamir region. (b) View of the valley of Sul-Iistyk river. (c) View of the cave entrance. (d) Stratigraphic NE profiles of the cave as of 2019.

**Table 1. Stratigraphy of the Istykskaya cave.**

| Stratigraphic unit | Depth (m) | Description | Archaeology | Absolute $^{14}$C dates (calibrated) |
|---|---|---|---|---|
| 1 | 0.0–0.4 | Gray silty sandy loam. | **Cultural layer 1**. The layer contains fire places, wooden pieces, broken bones and charcoal. | 4410–3981 years BP (GV 02109), this study |
| 2 | 0.45–0.53 | Brown to dark brown humified silty sandy loam. | **Cultural layer 2**. The layer contains numerous fire places and organic material (mostly animal dung). | 8635–8192 years BP (GV 02109), this study |
| 3 | 0.53–0.65 | Gray silty and structureless sandy loam. At the boundary between layers 3–4 a sterile layer is noted. | **Cultural layer 3**. The layer has yielded numerous animal bones, wood remains and lithic artifacts. | 14066–13351 years BP (NSKA 1622) and 13792–13605 (UGa 23052); see [13] |
| 4 | 0.65–0.85 | Well sorted gray river sand without inclusions. | Sterile | |

(OXCAL version 4.4). For cultural layer 3 two dates were previously obtained: 14066–13351 (NSKA 1622) and 13792–13605 (UGa 23052) [13].

The material described herein is deposited in the Archaeological Department of Institute of History, Archaeology and Ethnography (Dushanbe, Republic of Tajikistan). In the provided specimen numbers, the prefix IST refers to the collection from Istykskaya cave, 2018/2019 refers to the corresponding excavation season, and the following digit designates the cultural level.

The studied collection contains a diversity of bones of small passerine birds, which were not identified due to their fragmentary nature and generally uniform osteology. Only a small portion of passerine bones which display a characteristic morphology, were identified and are reported here. However, all bones of non-passeriform birds were identified and are considered in this work.

The general anatomical nomenclature in this work follows that of Baumel and Witmer [16], and the morphology of the quadratum follows Elzanowski and Stidham [17].

## Results: Systematic section

Order Anseriformes Wagler, 1831

Family Anatidae Leach, 1820

*Anas crecca* Linnaeus, 1758

**Material** (Fig 2C). Cranial fragment of right coracoid (specimen IST 2019-3-43); partial right scapula (specimen IST 2019-2-49); poorly preserved distal fragment of left tarsometatarsus (specimen IST 2018-2-3-su 7–15).

**Remarks.** The coracoid belongs to a small-sized teal, comparable in size with the modern *A. crecca* and *Spatula querquedula*, but distinctly smaller than *Sibirionetta formosa*. A distinct pit in the dorsal part of the sulcus musculi supracoracoidei (Fig 2, p), characteristic of *Spatula* ducks (and *S. querquedula* in particular), is absent. The absolute size is similar to the smallest modern specimens of *A. crecca* (reconstructed medial length is smaller than 34 mm), which are smaller than *S. querquedula* [18, 19]. The scapula (not figured) is also similar to the smallest specimens of *A. crecca* and thus is referred to this species.

The Eurasian teal is currently a rather common non-breeding and migrant species in Eastern Pamir [20].

Spatula clypeata (Linnaeus, 1758)

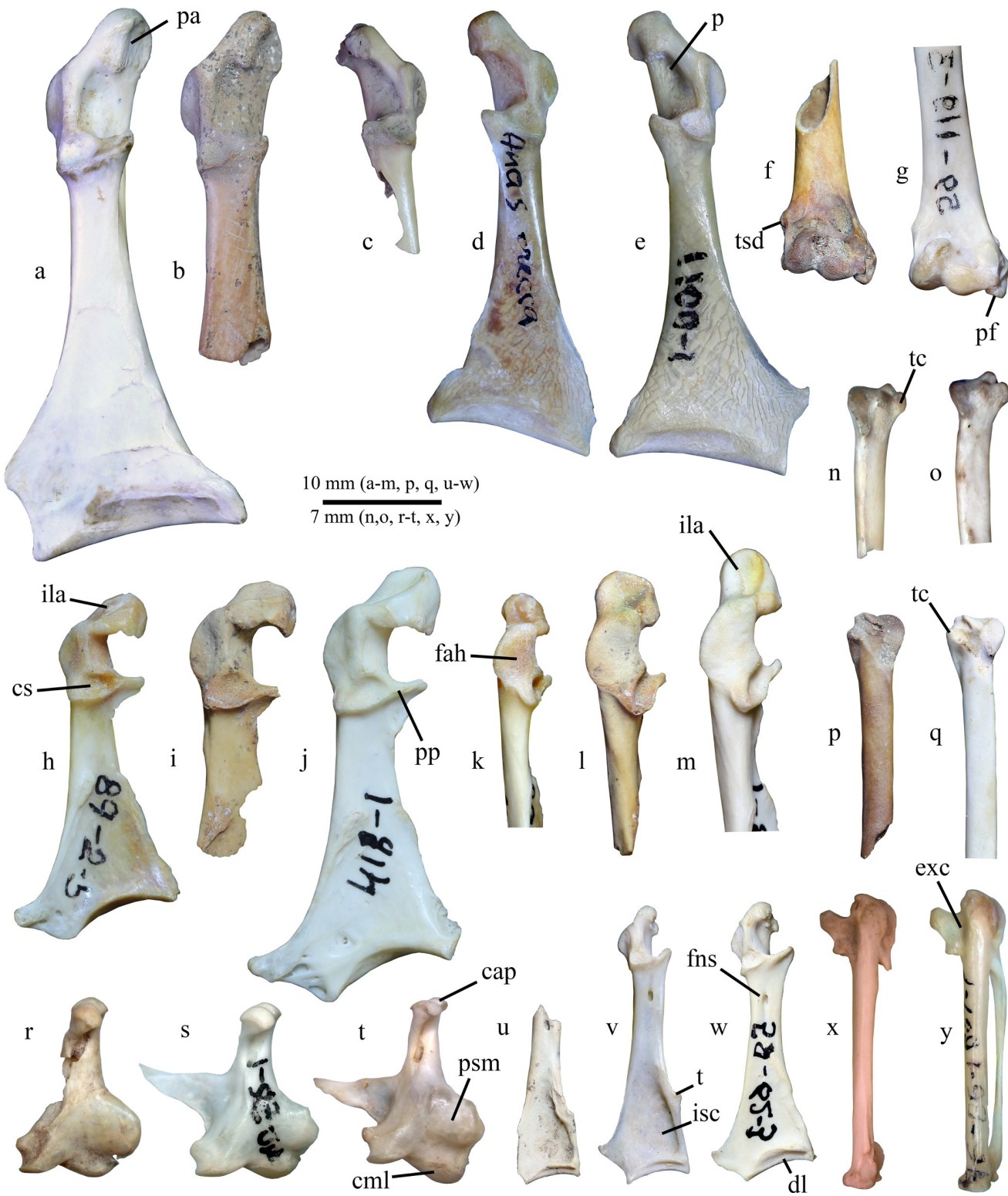

**Fig 2. Late Quaternary bones of non-passeriform birds from Istykskaya cave compared with selected modern taxa. a–e, h–m, u–w,** coracoids; **f, g,** humeri; **n, o, p, q,** ulnae; **r–t,** quadrati; **x, y,** carpometacarpi. **a–e, h–j, u–y,** dorsal views; **f, g,** cranial views; **k–m,** lateral views; **n, o, p, q,** ventral views; **r–t,** medial views: **a,** *Aythya fuligula* (modern, specimen PIN 40-68-1); **b,** *A. fuligula*, specimen IST 2019-3-69; **c,** *Anas crecca*, specimen IST 2019-3-43; **d,** *A. crecca* (modern; specimen PIN 40-23-1); **e,** *Spatula querquedula* (modern; specimen PIN 40-46-1); **f,** *Gallinula chloropus*, specimen IST 2019-2-49; **g,** *G. galleata* (modern, Cuba; specimen PIN 59-119-3); **h, k,** *Syrrhaptes paradoxus* (modern, specimen PIN 89-2-5); **i, l,** *S. tibetanus*, specimen IST 2019–2.4–36;

**j, m**, *Pterocles orientalis* (modern, specimen PIN 89-7-1); **n,** *Coturnix coturnix*, specimen IST 2019-3-42; **o,** *C. coturnix* (modern; specimen PIN 49-66-2); **p**, *Podiceps nigricollis/auritus*, specimen IST 2019-3-54; **q**, *P. nigricollis* (modern, specimen PIN 13-16-2); **r**, *Spatula clypeata*, specimen IST 2019–2.1–22; **s**, *S. clypeata* (modern, specimen PIN 40-52-1); **t**, *Mareca penelope* (modern; specimen PIN 40-17-1); **u**, *Rallus aquaticus*, specimen IST 2019-2-49; **v**, *R. aquaticus* (modern, specimen PIN 59-7-3); **w**, *Crex crex* (modern, specimen PIN 59-62-3); **x**, *Actitis hypoleucos* specimen IST 2019-2-49; **y**, A. hypoleucos (modern; specimen 76-26-1). Abbreviations: *cap*, capitula oticum et squamosum; *cml*, condylus mandibularis lateralis; *cs*, cotyla scapularis; *dl*, dorsal labrum of the facies articularis sternalis; *exc*, excavation at the base of the os metacarpale alulare; *fah*, facies articularis humeralis; *fns*, foramen n. supracoracoidei; *ila*, impressio ligamenti acrocoracohumeralis; *isc*, impressio m. sternocoracoidei; *p*, pit within the sulcus musculi supracoracoidei; *pa*, processus acrocoracoideus; *psm*, prominentia submeatica; *pf*, processus flexorius; *pp*, processus procoracoideus; *t*, tuberculum on the mediocranial angle of the impressio m. sternocoracoidei; *tc*, tuberculum carpale; *tsd*, tuberculum supracondylare dorsale.

**Material** (Fig 2R). Right quadratum (specimen IST 2019–2.1–22); proximal fragment of left tibiotarsus (specimen IST 2019-3-code).

**Remarks.** The quadratum plays a crucial role in the kinematics of the feeding apparatus of birds in general and filter-feeding ducks in particular [21–23], and thus it is expected that this bone would be diagnostic in ducks with different feeding specialization, of which the Northern Shoveler (*S. clypeata*) is one of the most striking examples [24]. The fossil specimen can be referred to Anatidae because it has a caudally shifted cotyla quadratojugalis, only two mandibular condyles and one-headed caput quadrati–characteristic features of the waterfowl quadratum, which is unusually built in comparison with that of most other birds [17, 23]. In size, the specimen is similar to *S. clypeata*, *Mareca penelope*, or *Aythya fuligula*. However, *Aythya* have widely-spaced capitula oticum et squamoum, which instead are closely-spaced in filter-feeding ducks [23]. *Aythya* further differs in the shape of prominentia submeatica, which in these ducks has a gradually sloping dorsal margin. The dorsal margin of the prominentia is more abrupt relative to the shaft of the bone in dabbling ducks (*Anas* s.l.). *Spatula clypeata* differs from other Palearctic dabbling ducks by a ventral displacement of the prominentia (together with cotyla quadratojugalis, which is located on it). In other dabbling duck, the ventral margin of the prominentia is shifted dorsally relative to the condylus mandibularis lateralis, whereas in *S. clypeata* it is positioned notably closer to it (Fig 2, psm). Additionally, the capitula oticum et squamosum are especially closely spaced in *S. clypeata* to form a ball-like articular surface, characteristic of specialized filter-feeders [23]. Also, in *M. penelope*, the condylus mandibularis lateralis is notably more ventrally prominent (Fig 2, cml).

The fragmentary tibiotarsus agrees with *S. clypeata* in size and is tentatively referred to this species.

The Northern Shoveler occurs in Pamir only during migrations and appears to be a relatively rare visitor of the highland territories [14, 25, 26].

*Aythya fuligula* (Linnaeus, 1758)

**Material** (Fig 2B). Omal fragment of left coracoid (specimen IST 2019-3-69).

**Remarks.** The coracoid can be referred to diving ducks because of the characteristic orientation of the plane of the processus acrocoracoideus that is more angled relative to the plane of the dorsal surface of the shaft than it is in dabbling ducks (see [27], character 45). In general size and proportions, the specimen agrees with *A. fuligula*. The coracoid of *A. fuligula* and *A. ferina* do overlap in overall size [19], but the fossil specimens agrees with the small-sized *A. fuligula* (greatest length, reconstructed given the size of the omal end, is no more than 47 mm) and is thus smaller than *A. ferina* (greatest length ~49–52 mm in most specimens [19]). The extremitas omalis is further somewhat more gracile than in *A. ferina*. In addition, the attribution to *A. fuligula* is indirectly supported by the notion that this species occurs at notably higher altitudes (up to ~3400 m [3]). The coracoid of *A. nyroca* is distinctly smaller and *A. marila* is distinctly larger. Similarly-sized ducks of the tribe Mergini have a notably more robust coracoid. The specimen displays several short cut-marks on the dorsal surface of the

shaft, indicating that the bird carcass was processed by humans. Cut-marks on the avian coracoid generally result from filleting rather than disarticulation of bird carcasses [28].

The Tufted Duck (*A. fuligula*) is currently a rather common (predominately autumn) migrant species in the Eastern Pamir [14, 20, 26].

Order Galliformes Temminck, 1820

Family Phasianidae Horsefield, 1821

*Coturnix coturnix* (Linnaeus, 1758)

**Material** (Fig 2O). A distal fragment of right ulna (specimen IST 2019-3-42).

**Remarks.** The bone displays a typical morphology of galliform birds; it has a distinctly protruding tuberculum carpale (Fig 2, tc) and a sharp ventrocaudal margin of the bone. In size, it corresponds to *Coturnix coturnix* and metrically strongly differs from other (larger) representatives of the Palearctic Phasianidae.

Presently, the Common Quail is absent in Eastern Pamir as breeding species, but the species breeds in Northern and Western Pamir (at elevations below 3000 m [14]). The Quail also occurs at Eastern Pamir on migrations [20, 25, 26, 29]. According to [3], the species now occurs globally at elevations up to 3600 m.

Order Podicipediformes Fürbringer, 1888

Family Podicipedidae Bonaparte, 1831

*Podiceps sp.* (*P. nigricollis/auritus*)

**Material** (Fig 2P). Distal fragment of left ulna (specimen IST 2019-3-54).

**Remarks.** The specimen can be referred to Podicipedidae due to characteristic proximally elongate base of the tuberculum carpale (Fig 2, tc) with a characteristic concavity on its ventral surface. In size, the ulna corresponds with the living *P. nigricollis* and *P. auritus*, which apparently cannot be distinguished based on isolated distal ulnae [30].

Either Horned (*P. auritus*) or Black-necked (*P. nigricollis*) Grebes are currently totally absent in Pamir, and can be found only in lowland Tajikistan during autumn migrations [20, 25, 29]. Globally, these grebes occur at elevations up to 3500 m [3].

Order Charadriiformes Huxley, 1867

Family Scolopacidae Rafinesque, 1815

*Actitis hypoleucos* (Linnaeus, 1758)

**Material** (Fig 2X). Left carpometacarpus (specimen IST 2019-2-49).

**Remarks.** The bone belongs to a small wader and is characterized by elongate slim proportions, no intermetacarpal process and non-roundish trochlea carpalis, which especially distinguishes it from similarly built carpometacarpi of Rallidae. In morphology and size, the bone is identical to that of *Actitis hypoleucos* and hence is referred to this modern species. Similarly-sized plovers (*Charadrius* spp.) have a notably more roundish ventral trochlea carpalis, notably thicker apex of the processus extensorius, poorly pronounced fossa supratrochlearis, and a longer distal symphysis. Sandpipers of the genus *Calidris* are morphologically very close to *Actitis*, and the species *C. alpina*, which today occurs in highlands of Pamir during migrations [14], is further very similar to the fossil bone in size. However, *Actitis* can be distinguished from *Calidris* by the presence of a well-defined excavation at the base of the os metacarpale alulare on its dorsal side (Fig 2, exc). This area is nearly flat in *Calidris* waders.

The Common Sandpiper is a common breeding species in Pamir, occurring at elevations up to 3800 m [14, 29]. Globally, the species occurs at mountains up to 4000 m [3].

Order Gruiformes Bonaparte, 1854

Family Rallidae Rafinesque, 1815

*Gallinula chloropus* (Linnaeus, 1758)

**Material** (Fig 2F). Distal fragment of right humerus, belonging to a subadult bird (specimen IST 2019-2-49), left ulna (specimen IST 2019-3-47.4), proximal fragment of left carpometacarpus (specimen IST 2019–2.3–32).

**Remarks.** The distal humerus belongs to a rather large rallid and is characterized by the following features characteristic of this family: a narrow distal end with distally protruding processus flexorius (Fig 2, pf), cranially elevated tuberculum supracondylare ventrale and low and proximally shifted tuberculum supracondylare dorsale (see comparisons in [31]). The specimen is roughly similar in size to *G. chloropus*, being distinctly smaller than *Fulica* and *Porphyrio*, but still larger than all other Palearctic rails. Interestingly, the specimen belonged to a remarkably large individual, comparable in size with *G. galeata cerceris* from Greater Antilles, which are the largest representative of the genus [32]. The porose surface of the distal end indicates that the specimen comes from a subadult individual. On the caudal surface of the fossa olecrani, there is an artificial notch, made by overextending the joint to disarticulate the wing [33]. The proximal carpometacarpus has a roundish trochlea carpalis and proximally oriented processus extensorius as in rails, it can also be referred to *G. chloropus* based on size. The ulna also agrees in size and morphology with *G. chloropus*.

The species rarely occurs in Pamir Mountains now and can be seen only during autumn migrations [14, 25]. According to del Hoyo [3], the species occurs at elevations up to 3000 m.

Rallus aquaticus Linnaeus, 1758

**Material** (Fig 2U). Caudal fragment of a left coracoid (specimen IST 2019-2-49).

**Remarks.** The specimen shows a characteristic morphology of Rallidae with trapezoid-shaped and dorsally excavated extremitas sternalis (see [31] for intrafamiliar comparisons). The overall length of the fossil coracoid, despite its fragmentary nature, can be reconstructed because the foramen n. supracoracoidei is preserved. In the length between the foramen and the caudal margin of the bone, the specimen is comparable with modern *Crex crex* and *Rallus aquaticus*. It is distinctly smaller than *Gallinula chloropus*. However, the proximomedial angle of the impressio m. sternocoracoidei is not developed in *Crex*, whereas in *Rallus* this area forms a distinct tubercle, which is also present in the fossil specimen (Fig 2, t). Additionally, in *Crex*, the shaft is thicker and its medial margin is concave [31], and the dorsal labrum of the facies articularis sternalis (Fig 2, dl) expands laterally, whereas the labrum is evenly narrow in *Rallus*.

The Water Rail globally occurs at altitudes no higher than 2000 m [3]. The species now inhabits only the foothills of the Pamir Mountains and is absent in highlands [14, 20].

Order Pterocliformes Huxley, 1868

Family Pteroclidae Bonaparte, 1831

*Syrrhaptes tibetanus* (Gould, 1850)

**Material** (Fig 2I and 2L). Cranial part of a right scapula (specimen IST 2019-3-50), cranial fragments of a left scapula and coracoid (specimens IST 2019–2.4–36).

**Remarks.** The coracoid displays a characteristic morphology of Pteroclidae and distinctly differs from the somewhat similar Charadriiformes by a very large and strictly dorsally oriented cotyla scapularis and reduced processus procoracoideus. Columbidae can easily be distinguished from Pteroclidae in that their coracoid has a smaller, non-excavated and laterally positioned cotyla scapularis (deeply concave, wide and centrally positioned in Pteroclidae), much wider and cranially oriented processus procoracoideus (thin and mostly medially oriented in Pteroclidae), straight and elongate shaft (it is notably shorter and gradually widens caudally in Pteroclidae), as well as notably less medially protruding processus acrocoracoideus. The specimen is morphologically similar to that of *Syrrhaptes paradoxus* and differs from *Pterocles orientalis* by a short impressio ligamenti acrocoracohumeralis, short and evenly dorsoventrally high facies articularis humeralis, and wider (craniocaudally larger) cotyla scapularis (Fig 2, cs, fah, ila). Most importantly, the processus acrocoracoideus is notably more cranially protruding in *Pterocles* than in the fossil and *S. paradoxus*. The fossil species however differs from *S. paradoxus* by a notably larger size, wider facies articularis humeralis, shorter impressio ligamenti acrocoracohumeralis, and the lack of extensive excavation beneath the facies articularis clavicularis. In its large size, the specimen corresponds with *S. tibetanus*, which is notably bigger than *S. paradoxus*. Morphological distinctiveness of *S. tibetanus* in comparison with *S. paradoxus* was already reported based on general morphology, which even resulted in treatment of the Tibetan Sandgrouse as a distinct genus, probably more closely related to other Pteroclidae than *S. paradoxus* [34, 35]. The structure of the coracoid, as revealed by the new material from Pamir, does not contradict such treatment.

Both scapulae (not shown) also agree in morphology with *S. paradoxus* and differ from *P. orientalis* by shorter and blunter acromion. They are however remarkably larger than *S. paradoxus* and hence are referred to *S. tibetanus*.

The Tibetan Sandgrouse currently inhabits highlands (more than 3000 m) of Eastern Pamir, which represent the western most part of its geographical range [3, 14, 26, 29, 35]. The find represents the first documentation of this species in the fossil record.

Order Passeriformes Linnaeus, 1758

Family Corvidae Leach, 1820

*Pyrrhocorax pyrrhocorax* (Linnaeus, 1758)

**Material** (Fig 3F). Cranial fragment of left coracoid (specimen IST 2019-2-26).

**Remarks.** The specimen has a characteristic morphology of perching birds (Passeriformes) and is referred to Corvidae because of its large size. Among Corvidae, the specimen agrees in size [36] with the largest specimens of *Corvus monedula* and *Pica pica*, as well as middle-sized specimens of *Pyrhocorax pyrrhocorax*. In *Pica pica*, the impressio ligamenti acrocoracohumeralis is notably shorter than in the fossil specimen. On the ventral ("cranial") surface of the processus acrocoracoideus, there is a depression, which is defined in *C. monedula*, but is poorly developed or absent in *Pyrrhocorax* [36]. This character supports the allocation of the bone to this species. Additionally, the apex of the processus acrocoracoideus points cranially in the fossil specimen and *P. pyrrhocorax*, whereas in *C. monedula* it is oriented more laterally.

The red-billed chough is presently a common species in the highlands of Pamir Mountains (up to 4500 m [14, 26, 37, 38]).

Family Motacillidae Horsfield, 1821

*Motacilla* ?*citreola* (Pallas, 1776)

**Material** (Fig 3A). Complete right coracoid (specimen IST 2019-3-74).

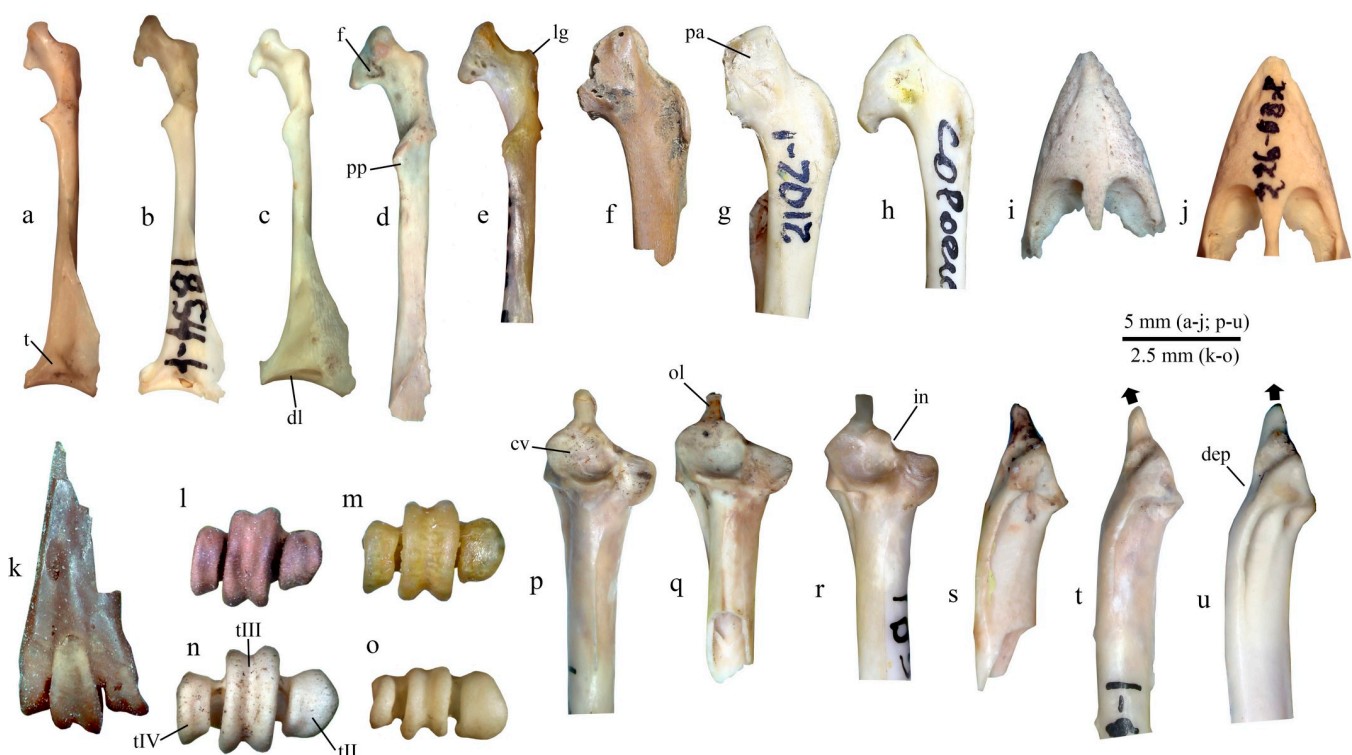

**Fig 3. Late Quaternary bones of perching birds (Passeriformes) from Istykskaya cave compared with selected modern taxa. a–h**, coracoids; **i, j**, rostra maxillae; **k–o**, tarsometatarsi; **p–u**, ulnae. **a–e, i–k**, dorsal views; **f–h, s–u**, ventral views; **l–o**, distal views; **p–r**, cranial views; **a**, *Motacilla ? citreola*, specimen IST 2019-3-74; **b**, *M. citreola* (modern; specimen PIN 168-3-1); **c**, *Acanthis flavirostris* (modern, specimen PIN 226-63-1); **d**, *Eremophila alpestris*, specimen IST 2018-2/3-SU7-15; **e, o**, *E. alpestris* (modern, specimen PIN 165-77-1); **f**, *Pyrrhocorax pyrrhocorax*, specimen IST 2019-2-26; **g**, *Pyrrhocorax pyrrhocorax* (modern, specimen PIN 244-65-1); **h**, *Corvus modenula* (modern, specimen PIN 244-68-1); **i**, *Leucosticte* sp., specimen IST 2019-3-42; **j**, *Leucosticte brandti* (modern, specimen PIN 226-68-2); **k, l, q, s**, *Monticola saxatilis*, specimen IST 2019–1.1–15 (**k, l**) and IST 2019-3-44 (**q, s**); **m**, *M. saxatilis* (modern, specimen PIN 185-186-1); **n, p, t**, *M. solitarius* (modern, specimen PIN 185-189-1); **r, u**, *Turdus atrogularis* (modern; specimen PIN 185-273-3). Abbreviations: *cv*, cotyla ventralis; *dep*, depression; *dl*, dorsal lobe of the facies articularis sternalis; *f*, foramen on the dorsal side of processus acrocoracoideus; *in*, incisura; *lg*, labrum glenoidale; *ol*, olecranon; *pa*, processus acrocoracoideus; *pp*, processus procoracoideus; *t*, tuberculum; *tI-tIII*, trochleae metatarsorum I-III. Arrow indicate the orientation of the olecranon.

**Remarks.** The complete coracoid belongs to a small passeriform bird and displays a combination of features which is characteristic of Motacillidae: the processus procoracoideus (Fig 3, pp) has a sharp apex and a concave cranial margin, and the dorsal lobe of the facies articularis sternalis is short, set apart from the angulus medialis, and there is a sub-triangular tubercle (Fig 3, t) located just cranial to this facies. In size, the specimen is similar to *M. citreola* and thus is preliminary referred to this species. It is however also possible that the bone belongs to small individuals of *M. cinerea*.

The Citrine Wagtail is a very common breeding species in the highlands of Pamir in contrast to Grey Wagtail (*M. cinerea*), which occurs in Eastern Pamir only during migrations [29, 38].

Family Fringillidae Leach, 1820

*Leucosticte nemoricola/brandti*

**Material** (Fig 3I). Rostra maxillarum (specimens IST 2019-3-42, 56), complete left carpometacarpus (specimen IST 2019-3-44).

**Remarks.** Beaks are probably most diagnostic elements of the passerine bird skeleton, sometimes allowing separation of close species [39]. The maxillae have a very broad base, as in

Passeridae and some Fringillidae, but they lack a ventral band in the lateral margin of the bone, which characterizes Passeridae. The rostra are much larger than in *Bucanetes mongolicus* and agree in size and morphology with *Leucosticte* finches. *Rhodopechys sanguineus* is a larger bird with more robust beak. The carpometacarpus is rather uniformly built in passerines, and the specimen from layer 3 is referred to *Leucosticte* preliminary, based on similar size and proportions.

The two species of *Leucosticte* are all-year residents of the highlands of Pamir Mountains [29, 40].

Family Alaudidae Vigors, 1825

*Eremophila alpestris* (Linnaeus, 1758)

**Material** (Fig 3D). Complete right humerus (IST 2018-2-21 SU 7X), right coracoid (IST 2018-2/3-SU7-15), proximal fragment of right humerus (IST 2019-3-67).

**Remarks.** Two humeri differ from all other (unidentified) paseriform humeri from the locality in the poor development of the dorsal tricipital fossa, as in Alaudidae (see [41–43] for differentiation from other families). They further have a well-marked notch in the proximal outline of the bone made by the incisura capitis, which is also characteristic of larks. In size and general morphology, the specimens are similar to *Eremophila*. *Alauda* larks have a more proximally protruding caput humeri relative to the tuberculum ventrale. Species of the genus *Calandrella*, and especially the Hume's Lark (*C. acutirostris*), which inhabits the region today, are remarkably smaller. The coracoid also displays a characteristic morphology of Alaudidae, mostly manifested in the pronounced cranial part of the labrum glenoidale and the presence of a pneumatic foramen on the dorsal side (see [43]).

The horned lark is a common species in Pamir, breeding at elevations up to 4500 m [14, 26, 37, 38].

Family Muscicapidae Fleming, 1822

*Monticola saxatilis* (Linnaeus, 1766)

**Material** (Fig 3K, 3L, 3Q and 3S). Distal fragment of right tarsometatarsus (specimen IST 2019–1.1–15); proximal fragment of left ulna (specimen IST 2019-3-44).

**Remarks.** The distal tarsometatarsus has an unusual configuration of the trochleae metatarsorum in distal view. In this specimen, the trochleae metatarsi II et IV are closely pressed to trochlea metatarsi III, and thus the distal end is very narrow, whereas it is usually much broader in Palearctic passerines. A similarly narrow distal end of tarsometatarsus is present in Sylviidae *sensu lato*, but the described specimen is notably larger than warblers and have a dorsoplantarly short trochlea metatarsi IV, whereas this trochlea is as high as trochlea metatarsi III in Sylviidae. In this morphology and size, the specimen is identical to *Monticola saxatilis* and hence is referred to this species here. *M. solitarius* and Turdidae have a more generalized wide distal tarsometatarsus.

The ulna belongs to a medium-sized passeriform bird, similar in size to *Turdus merula/atrogularis*, *Lanius excubitor*, *Sturnus vulgaris* and *Monticola* spp. The ulna of Sturnidae is characterized by a notably more robust olecranon, which is thin in other mentioned taxa. *Monticola* has a more caudally oriented olecranon, which is oriented more proximally in *Turdus* and *Lanius* (in ventral view). Additionally, *Turdus* differs from *Monticola* by the presence of a well-defined incisure in the dorsal margin of the cotyla ventralis and a concavity in the caudal margin of the bone in ventral view (at the level of the cotyla ventralis), and further by a non-convex proximoventral margin of the cotyla ventralis. Differentiation between *M. saxatilis* and *M. solitarius* may not be possible based on the preserved fragment of ulna, and we preliminary assign it to *M. saxatilis* based on the identification of the tarsometatarsus.

Nowadays the Rufous-tailed Rock Thrush breeds only at western Pamir at altitudes up to 3500 m, but still rarely occurs at higher altitudes of Eastern Pamir during autumn migrations, when it can be seen at elevations up to 4300 m [14, 38]. Among representatives of the genus *Monticola*, this species occurs at highest altitudes (globally up to 5000 m [3]).

## Discussion

The avian fauna from Istykskaya cave (Table 2) is the first fossil bird assemblage from Eurasian highlands (above 3000 m), and thus it contributes significantly to our knowledge of the history of Central Asian high mountain ecosystems, and bird communities in particular. Previously, only a couple of bird remains were described from somewhat less elevated areas of Tibet (see Introduction [4, 5]), and all other Asian fossil bird assemblages come from low mountain areas with elevation of approximately 1000 m or less [7, 44–49]. In Europe, several Quaternary avian faunas are known from less elevated areas in Alps and Caucasus (Drachenloch in Switzerland is the highest avian locality, with an altitude of 2427 m [44]). Furthermore, no fossil birds were previously described from Tajikistan. The new fauna is especially interesting because it originates from Eastern Pamir, an area with especially severe climate and environmental conditions (see above), and thus it sheds lights on the past history of these specific ecosystems with poorly known evolutionary history. It must be noted that there are no paleoelevation data for the Istyk river valley in the latest Pleistocene and early Holocene. Earlier geological estimates of the uplift rate of the Eastern Pamir during the late Quaternary ranged from 20 to 62 mm/year, and hence about 13 ka ago the valley must have been positioned some 260 to 850 m below the present level [50, 51]. However, the recent surface uplift rate of the north-eastern Pamir is estimated as only 1–5 mm/year [52], and such a slower rate would imply a significantly less uplift since the late Quaternary.

The composition of the bird fauna from Istykskaya cave indicates notably milder environmental conditions in Eastern Pamir during late Quaternary. The fauna contains a remarkable diversity of water birds, which is unexpected because presently the Eastern Pamir plateau is covered by high montane semi-deserts and only a small stream is present in the Valley of Istyk River. This diversity of waterbirds is especially notable as it includes forms with different ecological preferences. The presence of phytophagous dabbling ducks, zoophagous diving ducks

**Table 2. Occurrence of various bird taxa in different layers of Istykskaya cave.**

| taxon | cultural layers | | |
|---|---|---|---|
| | 3 (late Pleistocene) | 2 (early Holocene) | 1 (middle Holocene) |
| *Common Teal (Anas crecca)* | X | X | |
| *Northern Shoveler (Spatula clypeata)* | ? | X | |
| *Tufted Duck (Aythya fuligula)* | X | | |
| *Common Quail (Coturnix coturnix)* | X | | |
| *Grebe (Podiceps nigricollis/auritus)* | X | | |
| *Common Sandpiper (Actitis hypoleucos)* | | X | |
| *Common Moorhen (Gallinula chloropus)* | X | X | |
| *Water Rail (Rallus aquaticus)* | | X | |
| *Tibetan Sandgrouse (Syrrhaptes tibetanus)* | X | X | |
| *Cough (Pyrrhocorax pyrrhocorax)* | | X | |
| *?Citrine Wagtail (Motacilla ?citreola)* | X | | |
| *Mountain Finch (Leucosticte sp.)* | X | | |
| *Horned Lark (Eremophila alpestris)* | X | X | |
| *Common Rock Thrush (Monticola saxatilis)* | X | | X |

and grebes, as well as shorebirds and rails, indicate a relative abundance of both surface, benthic and shore resources and definitively points to the existence of a rather productive waterbody. A detailed study of distribution of various bird species at Pamir [40] emphasized the importance of local humidity, which correlates with the richness in vegetation and invertebrates. Indeed, increased water availability during the late Pleistocene through middle Holocene was recently shown from Karakul lake in northeastern Pamir [53, 54]. Lake levels much higher than today were identified for the period from ~19ka to 6.5 ka, with the highest level (~35 m above present) occurring at ~ 15 ka [53, 54]. Increased productivity of Karakul Lake

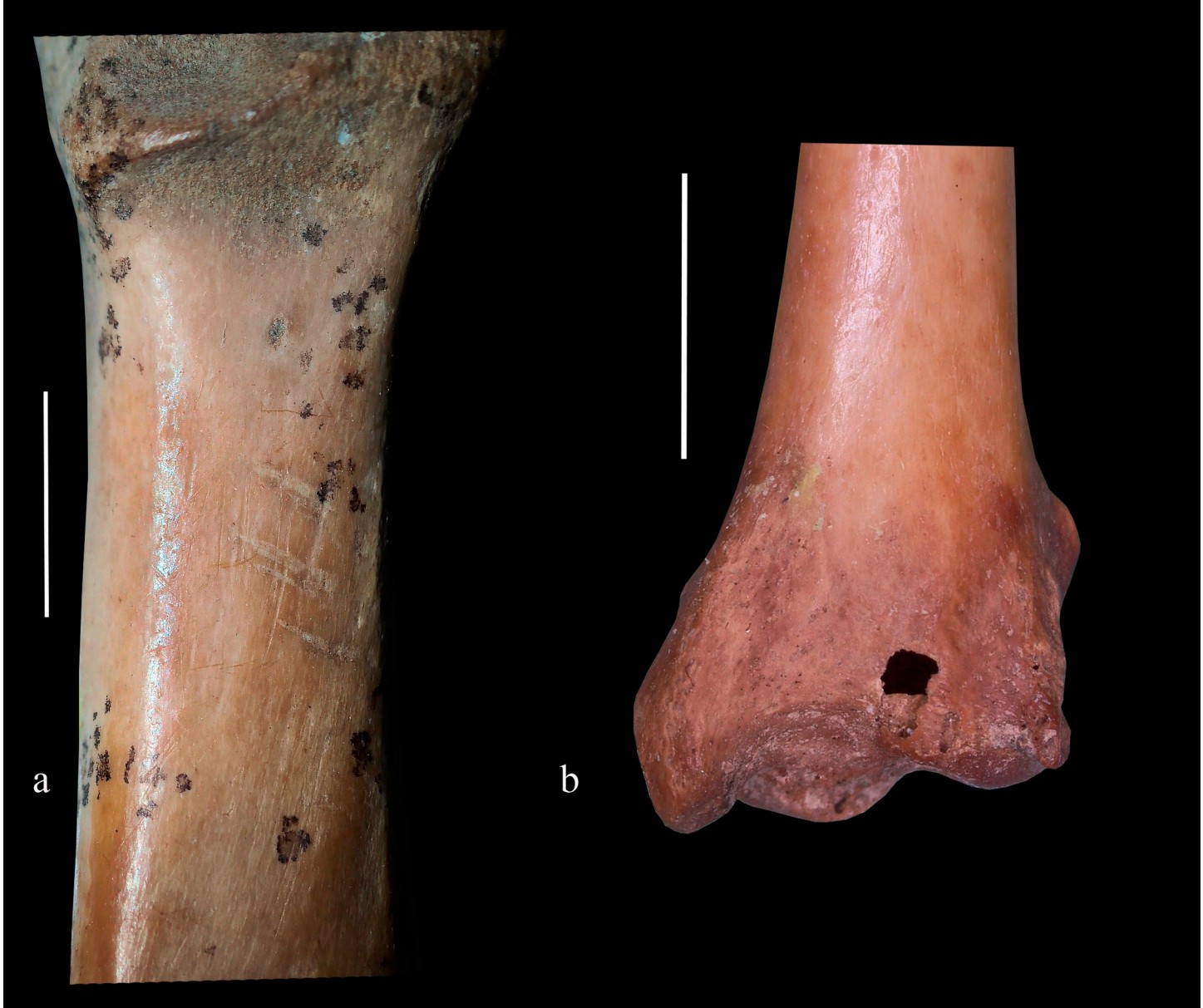

**Fig 4. Human-caused modifications (arrows) on avian bones from Istykskaya cave. a**, cut-marks on the dorsal surface of coracoid of *A. fuligula*, specimen IST 2019-3-69; **b,** humerus of *Gallinula chloropus* (specimen IST 2019-2-49), showing characteristic notch on the caudal surface of the fossa olecrani, made by overextending the joint to disarticulate the wing.

can also be traced for late Glacial to middle Holocene time interval [55]. Higher level of lakes at Eastern Pamir during late Glacial apparently resulted from decreased summer evaporation, and this implies a hydrological regime that is different from the present one [56]. Today, river valleys and lakes of Eastern Pamir are especially attractive for birds on migrations, and autumn migration in particular, which in this area starts as early as July [14]. The presence of Water Rail and *Podiceps nigricollis/auritus* grebe in the sediments of Istykskaya are especially remarkable, as these species were not recorded at such high altitudes globally [3]. It is thus may be concluded that productive waterbodies of Eastern Pamir attracted more lowland inhabitants during late Glacial and early-middle Holocene, including even some species that do not occur in this region now. Presently, *Aythya fuligula*, *Coturnix coturnix* and *Monticola* thrushes can be seen in Eastern Pamir only during autumn migration [14], but richer resources of the proposed waterbody might have attracter these species in other seasons as well.

Besides water birds, the avian assemblage from Istykskaya includes a diversity of specialized montane species, presently confined to highland areas of Central Asia. Especially notable is the find of the Tibetan Sandgrouse *Syrrhaptes tibetanus*, a specialized highland species of likely Tibetan origin [35], which was never reported before in the fossil record. Other bird taxa shared with the modern avian fauna of the Tibetan Plateau are *Pyrrhocorax pyrrhocorax*, *Eremophila alpestris*, *Motacilla citreola*, and *Leucosticte* sp. This assemblage shows that close zoogeographical affinities of the Eastern Pamir with the Tibetan Plateau [37] were already established during late Glacial. However, as the Eastern Pamir Plateau is still a largely isolated highland area, the exchange of alpine avian faunas with Tibet in the past might have only been possible via the Karokoram range [37]. The presence of above mentioned montane bird species further indicates the persistence of rather stable highland environments in the higher parts of the valleys during the late Quaternary, which agrees with palaeovegetation data [57].

Istykskaya cave further documents one of the earliest occurrences of Paleolithic humans in Eastern Pamir and Central Asian highlands in general [13]. The occurrence of humans in this area around 13.5 ka may be linked with the above-documented climatic changes and greater humidity in particular [58]. A previous analysis has shown that Common Quail *Coturnix coturnix* is the avian species that is most frequently found in Palaeolithic sites across Eurasia [9]. It is notable that the quail is also present in the deep level of Istykskaya cave together with stone artefacts, although this species is only a rare migrant in Eastern Pamir at present time. Although the only quail bone from Istykskaya does not bear traces of human impact, this species still could have been consumed by humans, because several bones of waterfowl from the cave display cutmarks (Fig 4; see also descriptions above). This means that at least some bird remains (particularly, those of larger species) accumulated at the site as a result of human activity. Smaller passeriform birds were most likely brought to the cave by predators or died there naturally.

## Acknowledgments

The authors thanks all the participants of the field work at Istykskaya cave and three anonymous reviewers for valuable comments.

## Author Contributions

**Conceptualization:** Nikita Zelenkov, Svetlana V. Shnaider.

**Data curation:** Nuritdin Sayfulloev, Svetlana V. Shnaider.

**Formal analysis:** Nikita Zelenkov.

**Funding acquisition:** Nikita Zelenkov, Svetlana V. Shnaider.

**Investigation:** Nikita Zelenkov, Svetlana V. Shnaider.

**Methodology:** Svetlana V. Shnaider.

**Project administration:** Nuritdin Sayfulloev.

**Resources:** Nuritdin Sayfulloev, Svetlana V. Shnaider.

**Supervision:** Nuritdin Sayfulloev.

**Writing – original draft:** Nikita Zelenkov.

**Writing – review & editing:** Nikita Zelenkov.

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
