## [Decision Letter · Decision Letter 0]

30 Jun 2021

PONE-D-21-16734

Fossil birds from the Roof of the World: the first avian fauna from High Asia, and its implications for late Quaternary environments in Eastern Pamir

PLOS ONE

Dear Dr. Zelenkov,

Thank you for submitting your manuscript to PLOS ONE. After careful consideration, we believe it has merit although it needs some corrections to fully meet the PLOS ONE publication criteria as they currently stand. Therefore, we invite you to submit a revised version of the manuscript that addresses the points raised by the three reviewers during the review process. When you do, please pay special attention to the reviewer's questions 1.

We look forward to receiving your revised manuscript.

Kind regards,

Claudia Patricia Tambussi, Ph.D.

Academic Editor

PLOS ONE

Journal Requirements:

2. In your Methods section, please provide additional location information of the study site, including geographic coordinates for the data set if available.

5. We note that Figure  in your submission contain map/satellite images which may be copyrighted. All PLOS content is published under the Creative Commons Attribution License (CC BY 4.0), which means that the manuscript, images, and Supporting Information files will be freely available online, and any third party is permitted to access, download, copy, distribute, and use these materials in any way, even commercially, with proper attribution. For these reasons, we cannot publish previously copyrighted maps or satellite images created using proprietary data, such as Google software (Google Maps, Street View, and Earth). For more information, see our copyright guidelines: http://journals.plos.org/plosone/s/licenses-and-copyright.

5.1.    You may seek permission from the original copyright holder of Figure 1 to publish the content specifically under the CC BY 4.0 license. 

5.2.    If you are unable to obtain permission from the original copyright holder to publish these figures under the CC BY 4.0 license or if the copyright holder’s requirements are incompatible with the CC BY 4.0 license, please either i) remove the figure or ii) supply a replacement figure that complies with the CC BY 4.0 license. Please check copyright information on all replacement figures and update the figure caption with source information. If applicable, please specify in the figure caption text when a figure is similar but not identical to the original image and is therefore for illustrative purposes only.

Reviewers' comments:

Reviewer's Responses to Questions

**Comments to the Author**

1. Is the manuscript technically sound, and do the data support the conclusions?

Reviewer #1: Yes

Reviewer #2: Yes

Reviewer #3: Yes

2. Has the statistical analysis been performed appropriately and rigorously? 

Reviewer #1: N/A

Reviewer #2: N/A

Reviewer #3: N/A

3. Have the authors made all data underlying the findings in their manuscript fully available?

Reviewer #1: Yes

Reviewer #2: Yes

Reviewer #3: Yes

4. Is the manuscript presented in an intelligible fashion and written in standard English?

Reviewer #1: Yes

Reviewer #2: Yes

Reviewer #3: Yes

5. Review Comments to the Author

**Reviewer #1: **This is an interesting manuscript, which adds novel data on the avifaunas of Central Asia. I have only various minor comments you may wish to consider:

- lines 278 ff: For the coracoid of the Pteroclidae you characters distinguishing it from the coracoid of the Charadriiformes. However, wouldn’t it be more important to indicate how it differs from the coracoid of the Columbidae? I think the coracoids of the Pteroclidae and Columbidae are much more similar than those of the Pteroclidae and Charadriiformes.

A large-sized pigeon species that occurs in the Pamir region is the Snow pigeon, Columba leuconota, so that a differentiation from columbids would be needed.

- Line 362 ff: Concerning the identification of the lark: What about Calandrella acutirostris, which also occurs in the Pamir region?

-line 407: “first fossil bird assemblage from Eurasian highlands (above 3000 m)“ – are you sure? Are there no bird fossils from caves in the Alps (I am just asking and do not known myself)?

- Fig. 1: The style of the lettering of this figure does not correspond to that of the other figures

Minor comments:

- in the abstract, it would be good to indicate more exactly the geographic location of the Pamir plateau (i.e., that it is in eastern Tajikistan)

- abstract, line 21: "partly common with" reads awkward to me. Maybe, "which in part are shared with"?

- abstract, line 30: I think this should be "a few", not "few", since you intend to say that there are "some" species (rather than "not many species")

- line 51: "up to 2700 above sea level" - do you really mean "up to" here, or shouldn't that rather be something like "above" or "starting from"? I think the lower rather than the upper boundary should be given here.

- line 60: "by cold mountain steppes and deserts" (delete "a" before cold)

- line 65: here and elsewhere: "high altitude" may perhaps sound better than "high elevation" (elevation is, however , not wrong, so you may leave it)

- line 66: "represent the first evidence" sounds awkward. Maybe "for the first time yield insights into"

- line 75: excavated a total area of (instead of "who totally excavated an area")

- line 77: "and further excavated"

- "Table 2: Is "lithic artifacts" a correct term or should this be "stone artifacts"?

- line 95: “is deposited in” or “was transferred to”(not “referred in”)

- line 100: “diagnostibility“ is no existing word. Maybe “diagnosability” “low amount of diagnostic features”?

- line 166 and elsewhere: “capitula oticum et squamosum” (not “capituli”)

- line 175 and elsewhere: “fragment of omal extremity of left coracoid” (not “cranial fragment of left coracoid”)

- line 184: “is additionally indicated”. Better start sentence with “In addition, the…”

-line 189: “of bird carcasses” (plural)

- line 194: add author name after Phasianidae (since author names are also provided elsewhere)

- line 246: “similar in size to” (not “with”)

- line 339: “procoracoideus” (delete second “i”)

- line 350: “Rostrum maxillae” (if singular) or “Rostra maxillarum” (if plural, which I think is meant here)

- line 381: what is meant with “the trochlea intertarsorum”? Trochleae metatarsorum?

- line 385: “specimen is” or “specimens are”

- line 389: “similar in size to thrushes Turdus merula/atrogularis, Lanius excubitor, Sturnus…” It is misleading to add “thrushes” here, since most of the species you list are no thrushes

- line 397: “may not be possible based on the preserved fragment, but we preliminary assign it to M. saxatilis based on more precise identification of the other specimen” – this reads a bit confusing and I suggest to write “based on the preserved fragment of the ulna”

- line 413: “an area with”

-line 425: “Lake levels” (delete “the” at start of sentence)

- line 428: “Higher lake levels at Eastern Pamir” – I am not sure what exactly is meant with this. It reads, as if you want to indicate a higher water level of the lakes, but I assume you mean something like “lakes at higher altitude levels in the Eastern Pamir”

- line 432: “nigricollis/auritus grebe” – I suggest to add the genus here

- line 435: “ioncluding even some species that do not occur…”

- line 453: “with the above-documented”

- line 455: “increase in moisture availability” sounds awkward. Better “greater humidity”?line - line 454: “has shown”

**Reviewer #2:** This is a straight-forward description of new avian fossils of some interest to the paleornithological community. The writing is relatively clear and concise and most of the figures are of good quality. My comments are largely restricted to some suggestions for minor edits and additions that would improve this manuscript.

Comments:

Title: There is no need for a comma after “Asia”

Table 1: The addition of common names (e.g., Quail, grebe, etc…) would be useful to the reader. Also, some indication of the relative age of “cultural layers 1, 2, 3 would be helpful. Finally, this table seems out of place and should likely be moved down to a spot after the materials and Methods as you have not introduced the reader to most of these specimens/taxa yet. This table is a “Result” of your study.

Table 2: charcoal is both singular and plural, depending on context (delete the “s”)

Line 89: Golden Valley (capitalize?)

Line 101: “characteristic morphology, were identified and are reported…”

Line 187: “cut-marks”

Line 260: “trapezoid-shaped”

Line 270 & elsewhere throughout the entire manuscript: The use of capitalization for common/English bird names is inconsistent (should be Water Rail). All English bird names should be capitalized (e.g., Tibetan Sandgrouse on line 296). Note, that the hyphenated parts of species name modifiers are not capitalized (e.g., Red-billed Chough). Common names are not capitalized when referring to a group in a general way unless they are the first word in a sentence (e.g., Hawks eat thrushes.). Please check the entire manuscript and correct as I did not but do see lines: 331, 345, 374, 399, 432.

Lines 297-298: You state that this is the first fossil record of Tibetan Sandgrouse but above on lines 290 you mention that this species has not always been considered distinct. Some mention of whether it is possible that other material, previously referred to the group could represent this taxon might be appropriate.

Line 385: I assume you mean “Sylviidae sensu lato”. Please write out in full (and properly italicize) or clarify.

Line 411: suggest change to “with elevations of approximately 1000 m or less”

Lines 406-413: This discussion would greatly benefit from the addition of data on the paleoelevation of the site during the times of deposition. What was the altitude of the cave from 4-13ka?

Line 412-413: this statement about the study being “especially important” because “of especially severe climate” is awkward and unclear as to your meaning. Please explain why this study is especially important and why it matters that the area is today characterized by extreme climate. Does it matter that the areas’ climate is extreme today when the climate was much less extreme during the time of deposition of the fossils?

Line 432: Write out the full names of these species (nigrocollis/auritus)

Line 433: “at such high altitudes”

Line 448: The Karkorum Range should be labeled on your map (Figure 1), as should many other features (see comment on Figure 1 below).

Line 454: “has shown”

Line 459: “waterfowl from the cave display cut-marks (Fig.4…”

Figure 1: This figure needs to be revised. The globe at the top is blurry and is essentially pointless at this scale. I assume the “red area” is the study area we see at a adjacent to it but this is not clear. Panel A needs the addition of labels for modern political boundaries, mountain ranges, paleoboundaries of the lake if available, etc… Panel A lacks information that the reader will be looking for to orient themselves based on the mention of physiographic features in your text. Panels B and C are blurry.

**Reviewer #3**: This is the first account of the montane avifaunas, from the latest Pleistocene through to the middle Holocene, of the Eastern Pamirs; the composition of the assemblages argues for milder climatic conditions at the time compared to the present. The manuscript is very well written and the conclusions are straightforward. I only have minor suggestions that might improve readability:

- p. 2 line 27. delete “further”

-p. 3. I recommend deleting the first sentence of the introduction as it lacks the context that makes it meaningful – as it stands it reads a bit vague. Alternatively, it could be moved towards the end of the introduction.

- p.3 line 54. The “However” does not seem to follow from the previous sentence, and it does not read like these two sentences are linked.

p. 4 line 73, add “the” before SW. Also p. 8, line 138 before modern; p. 13, line 279 before somewhat; p. 18 line 414 before bird

p. 4 line 75 delete “totally”.

p. 6, line 101, add “a” before characteristic. 100, replace with generally not diagnostic.

p. 8 line 143, mention that the scapula is not figured.

p. 12 line 249, replace “unfinished” with striated, porose, or similar.

p. 17, l 382. trochleae metatarsorum

I would also recommend using capitals for the common names of birds. I note that several abbreviations are used (e.g., lig.) and these should either be spelled out (at least at first mention) or included under “Abbreviations”.

6. PLOS authors have the option to publish the peer review history of their article (what does this mean?). If published, this will include your full peer review and any attached files.

Reviewer #1: No

Reviewer #2: No

Reviewer #3: No

---

## [Author Response · Author response to Decision Letter 0]

9 Sep 2021

Response to Reviewers

Dear Editors,

We are very grateful to the reviewer and editors for comments and suggestions on our manuscript. We have implemented all the requested changes, and our replies are given below in red.

Sincerely,

The authors

We have reviewed our reference list. No retracted papers are cited. We have added a couple of references related to paleo elevations, as requested by one of the reviewers. 

 checked. 

2. In your Methods section, please provide additional location information of the study site, including geographic coordinates for the data set if available.

Done. 

 We have added such information. 

 ORCID ID validated for corresponding author. 

5. We note that Figure in your submission contain map/satellite images which may be copyrighted. All PLOS content is published under the Creative Commons Attribution License (CC BY 4.0), which means that the manuscript, images, and Supporting Information files will be freely available online, and any third party is permitted to access, download, copy, distribute, and use these materials in any way, even commercially, with proper attribution. For these reasons, we cannot publish previously copyrighted maps or satellite images created using proprietary data, such as Google software (Google Maps, Street View, and Earth). For more information, see our copyright guidelines: http://journals.plos.org/plosone/s/licenses-and-copyright.

We have changed a figure, now providing an artist’s physical map of the region, and attach a license for this artwork. 

5.1. You may seek permission from the original copyright holder of Figure 1 to publish the content specifically under the CC BY 4.0 license. 

 Done

5.2. If you are unable to obtain permission from the original copyright holder to publish these figures under the CC BY 4.0 license or if the copyright holder’s requirements are incompatible with the CC BY 4.0 license, please either i) remove the figure or ii) supply a replacement figure that complies with the CC BY 4.0 license. Please check copyright information on all replacement figures and update the figure caption with source information. If applicable, please specify in the figure caption text when a figure is similar but not identical to the original image and is therefore for illustrative purposes only.

Reviewer #1: This is an interesting manuscript, which adds novel data on the avifaunas of Central Asia. I have only various minor comments you may wish to consider:

- lines 278 ff: For the coracoid of the Pteroclidae you characters distinguishing it from the coracoid of the Charadriiformes. However, wouldn’t it be more important to indicate how it differs from the coracoid of the Columbidae? I think the coracoids of the Pteroclidae and Columbidae are much more similar than those of the Pteroclidae and Charadriiformes.

A large-sized pigeon species that occurs in the Pamir region is the Snow pigeon, Columba leuconota, so that a differentiation from columbids would be needed.

Thank you for this comment. Indeed, Pteroclidae are superficially similar to Columbidae, though still differ significantly in osteology. We have added a comparison. 

- Line 362 ff: Concerning the identification of the lark: What about Calandrella acutirostris, which also occurs in the Pamir region?

We have added a comment on this, Calandrella are notably smaller larks. 

-line 407: “first fossil bird assemblage from Eurasian highlands (above 3000 m)“ – are you sure? Are there no bird fossils from caves in the Alps (I am just asking and do not known myself)?

Thank you for this note, we have somewhat clarified this – indeed, several sites are from altitudes above 2000 m in Europe, but none were reported from above 2500 m. 

- Fig. 1: The style of the lettering of this figure does not correspond to that of the other figures

Corrected. 

Minor comments:

- in the abstract, it would be good to indicate more exactly the geographic location of the Pamir plateau (i.e., that it is in eastern Tajikistan)

- abstract, line 21: "partly common with" reads awkward to me. Maybe, "which in part are shared with"?

- abstract, line 30: I think this should be "a few", not "few", since you intend to say that there are "some" species (rather than "not many species")

- line 51: "up to 2700 above sea level" - do you really mean "up to" here, or shouldn't that rather be something like "above" or "starting from"? I think the lower rather than the upper boundary should be given here.

- line 60: "by cold mountain steppes and deserts" (delete "a" before cold)

- line 65: here and elsewhere: "high altitude" may perhaps sound better than "high elevation" (elevation is, however , not wrong, so you may leave it)

- line 66: "represent the first evidence" sounds awkward. Maybe "for the first time yield insights into"

- line 75: excavated a total area of (instead of "who totally excavated an area")

- line 77: "and further excavated"

- "Table 2: Is "lithic artifacts" a correct term or should this be "stone artifacts"?

- line 95: “is deposited in” or “was transferred to”(not “referred in”)

- line 100: “diagnostibility“ is no existing word. Maybe “diagnosability” “low amount of diagnostic features”?

- line 166 and elsewhere: “capitula oticum et squamosum” (not “capituli”)

- line 175 and elsewhere: “fragment of omal extremity of left coracoid” (not “cranial fragment of left coracoid”)

- line 184: “is additionally indicated”. Better start sentence with “In addition, the…”

-line 189: “of bird carcasses” (plural)

- line 194: add author name after Phasianidae (since author names are also provided elsewhere)

- line 246: “similar in size to” (not “with”)

- line 339: “procoracoideus” (delete second “i”)

- line 350: “Rostrum maxillae” (if singular) or “Rostra maxillarum” (if plural, which I think is meant here)

- line 381: what is meant with “the trochlea intertarsorum”? Trochleae metatarsorum?

- line 385: “specimen is” or “specimens are”

- line 389: “similar in size to thrushes Turdus merula/atrogularis, Lanius excubitor, Sturnus…” It is misleading to add “thrushes” here, since most of the species you list are no thrushes

- line 397: “may not be possible based on the preserved fragment, but we preliminary assign it to M. saxatilis based on more precise identification of the other specimen” – this reads a bit confusing and I suggest to write “based on the preserved fragment of the ulna”

- line 413: “an area with”

-line 425: “Lake levels” (delete “the” at start of sentence)

- line 428: “Higher lake levels at Eastern Pamir” – I am not sure what exactly is meant with this. It reads, as if you want to indicate a higher water level of the lakes, but I assume you mean something like “lakes at higher altitude levels in the Eastern Pamir”

- line 432: “nigricollis/auritus grebe” – I suggest to add the genus here

- line 435: “ioncluding even some species that do not occur…”

- line 453: “with the above-documented”

- line 455: “increase in moisture availability” sounds awkward. Better “greater humidity”?line - line 454: “has shown”

Thank you, we have implemented all requested changes. We used “omal fragment” instead of suggested “fragment of omal extremety” for coracoids, because described fragments include not only extremity but additionally a part of the shaft. 

Reviewer #2: This is a straight-forward description of new avian fossils of some interest to the paleornithological community. The writing is relatively clear and concise and most of the figures are of good quality. My comments are largely restricted to some suggestions for minor edits and additions that would improve this manuscript.

Comments:

Title: There is no need for a comma after “Asia”

Table 1: The addition of common names (e.g., Quail, grebe, etc…) would be useful to the reader. Also, some indication of the relative age of “cultural layers 1, 2, 3 would be helpful. Finally, this table seems out of place and should likely be moved down to a spot after the materials and Methods as you have not introduced the reader to most of these specimens/taxa yet. This table is a “Result” of your study.

Table 2: charcoal is both singular and plural, depending on context (delete the “s”)

Line 89: Golden Valley (capitalize?)

Line 101: “characteristic morphology, were identified and are reported…”

Line 187: “cut-marks”

Line 260: “trapezoid-shaped”

Line 270 & elsewhere throughout the entire manuscript: The use of capitalization for common/English bird names is inconsistent (should be Water Rail). All English bird names should be capitalized (e.g., Tibetan Sandgrouse on line 296). Note, that the hyphenated parts of species name modifiers are not capitalized (e.g., Red-billed Chough). Common names are not capitalized when referring to a group in a general way unless they are the first word in a sentence (e.g., Hawks eat thrushes.). Please check the entire manuscript and correct as I did not but do see lines: 331, 345, 374, 399, 432.

Lines 297-298: You state that this is the first fossil record of Tibetan Sandgrouse but above on lines 290 you mention that this species has not always been considered distinct. Some mention of whether it is possible that other material, previously referred to the group could represent this taxon might be appropriate.

Line 385: I assume you mean “Sylviidae sensu lato”. Please write out in full (and properly italicize) or clarify.

Line 411: suggest change to “with elevations of approximately 1000 m or less”

Lines 406-413: This discussion would greatly benefit from the addition of data on the paleoelevation of the site during the times of deposition. What was the altitude of the cave from 4-13ka?

This is a really important notion; thank you. We have added an estimate of Pamir uplift during last 13 ka. 

Line 412-413: this statement about the study being “especially important” because “of especially severe climate” is awkward and unclear as to your meaning. Please explain why this study is especially important and why it matters that the area is today characterized by extreme climate. Does it matter that the areas’ climate is extreme today when the climate was much less extreme during the time of deposition of the fossils?

We have rephrased this. 

Line 432: Write out the full names of these species (nigrocollis/auritus)

Line 433: “at such high altitudes”

Line 448: The Karkorum Range should be labeled on your map (Figure 1), as should many other features (see comment on Figure 1 below).

Line 454: “has shown”

Line 459: “waterfowl from the cave display cut-marks (Fig.4…”

Figure 1: This figure needs to be revised. The globe at the top is blurry and is essentially pointless at this scale. I assume the “red area” is the study area we see at a adjacent to it but this is not clear. Panel A needs the addition of labels for modern political boundaries, mountain ranges, paleoboundaries of the lake if available, etc… Panel A lacks information that the reader will be looking for to orient themselves based on the mention of physiographic features in your text. Panels B and C are blurry.

Thanks so much for the comments, we have implemented requested changes. The Tibetan Sandgrouse was not always treated within the genus Syrrhaptes (sometimes was considered to represent a distinct genus), but it was always considered as a separate species. We have changed images of the Figure 1, and have added labels on the map. 

Reviewer #3: This is the first account of the montane avifaunas, from the latest Pleistocene through to the middle Holocene, of the Eastern Pamirs; the composition of the assemblages argues for milder climatic conditions at the time compared to the present. The manuscript is very well written and the conclusions are straightforward. I only have minor suggestions that might improve readability:

- p. 2 line 27. delete “further”

-p. 3. I recommend deleting the first sentence of the introduction as it lacks the context that makes it meaningful – as it stands it reads a bit vague. Alternatively, it could be moved towards the end of the introduction.

- p.3 line 54. The “However” does not seem to follow from the previous sentence, and it does not read like these two sentences are linked.

p. 4 line 73, add “the” before SW. Also p. 8, line 138 before modern; p. 13, line 279 before somewhat; p. 18 line 414 before bird

p. 4 line 75 delete “totally”.

p. 6, line 101, add “a” before characteristic. 100, replace with generally not diagnostic.

p. 8 line 143, mention that the scapula is not figured.

p. 12 line 249, replace “unfinished” with striated, porose, or similar.

p. 17, l 382. trochleae metatarsorum

I would also recommend using capitals for the common names of birds. I note that several abbreviations are used (e.g., lig.) and these should either be spelled out (at least at first mention) or included under “Abbreviations”.

Thank you, we have corrected all of these.

---

## [Editor Report · Decision Letter 1]

14 Oct 2021

Fossil birds from the Roof of the World: the first avian fauna from High Asia, and its implications for late Quaternary environments in Eastern Pamir

PONE-D-21-16734R1

Dear Dr. Zelenkov,

We’re pleased to inform you that your manuscript has been judged scientifically suitable for publication and will be formally accepted for publication once it meets all outstanding technical requirements.

Kind regards,

Claudia Patricia Tambussi, Ph.D.

Academic Editor

PLOS ONE
---

## [Editor Report · Acceptance letter]

18 Oct 2021

PONE-D-21-16734R1 

Fossil birds from the Roof of the World: the first avian fauna from High Asia and its implications for late Quaternary environments in Eastern Pamir 

Dear Dr. Zelenkov:

I'm pleased to inform you that your manuscript has been deemed suitable for publication in PLOS ONE. Congratulations! Your manuscript is now with our production department. 

Kind regards, 

on behalf of

Dr. Claudia Patricia Tambussi 

Academic Editor

PLOS ONE